# Log-Normal Multiplicative Dynamics for Stable Low-Precision Deep Learning

## Abstract

We design a new algorithm for stable low-precision deep learning motivated by the robustness of biological neural networks. It is well known that the stationary distribution of spine sizes follows a log-normal distribution and arises from noisy multiplicative dynamics. Building on these synaptic fluctuations that underlie neural computation, we propose the Log-normal Multiplicative Dynamics (LMD) algorithm for stable learning under low-precision computation. The method is derived by using variational training with a log-normal posterior distribution over the weights. LMD is a multiplicative weights update method that overcomes the scalability challenges seen in other multiplicative updates. We show empirically that LMD can learn stably under low-precision matrix multiplications during forward passes. It also gives accurate results for training-from-scratch for Vision Transformer and GPT-2 scale architectures. These results suggest that multiplicative dynamics, a biological feature, can maintain performance under low-precision computation, a promising direction for future energy-efficient hardware.

## 1 Introduction

Biological synapses exhibit continuous dynamic activity accompanied by noise (Choquet & Triller, 2013). The distribution of the synaptic spine size has been observed to follow a log-normal distribution, which is thought to arise from noisy multiplicative dynamics (Loewenstein et al., 2011). In addition to exhibiting activity-dependent plasticity, synaptic spines also show spontaneous fluctuations, with their dynamics considered to be approximately proportional to the spine size (Loewenstein et al., 2011; Kasai et al., 2021). Such dynamics are often regarded as a cause of unreliable or uncertain information transmission, but their importance in neural computation is being increasingly recognized (Seung, 2003; Bartol et al., 2015; Kappel et al., 2015; Aitchison et al., 2021).

These noisy multiplicative dynamics could help artificial neural networks (ANNs) tolerate imprecise computation when using low-precision data formats. Multiplicative fluctuations align with data formats that prioritize dynamic range at limited bit widths (Bernstein et al., 2020). This alignment is promising for energy-efficient dedicated hardware for inference and training (Zhao et al., 2022; Haghi et al., 2024). Among such formats, the Microscaling (MX) data format is a low-precision scheme that prioritizes dynamic range (Rouhani et al., 2023a). It has been actively considered for LLM deployments (Rouhani et al., 2023c; Verrilli, 2024; Tseng et al., 2025; Ramani et al., 2025).

To exploit the dynamics of biological synapses for ANNs, both the weight updates and the noise injection for the spontaneous fluctuations need to be multiplicative. However, contemporary ANN training employ only one of these two mechanisms. Multiplicative weight updates (MWUs) have been explored for training deep ANNs via the Madam optimizer (Bernstein et al., 2020), but noise injection has not been addressed. Furthermore, MWUs suffer from scalability issues and have not yet achieved successful training of large networks from scratch, such as Vision Transformers (ViT) (Dosovitskiy et al., 2021) or GPT-2 (Radford et al., 2019). In case of noise injection, Bayesian neural networks assume a distribution over their weights and perform sampling through noise injection into the weights (Graves, 2011; Blundell et al., 2015; Khan et al., 2018). However, because the sampling noise is not inherently proportional to the weight magnitude, it is insufficient to reproduce the fluctuations observed in biological synapses. Previous works use multiplicative noise proportional to the weight norm, but still use additive gradient updates (Wen et al., 2018; Bisla et al., 2022;

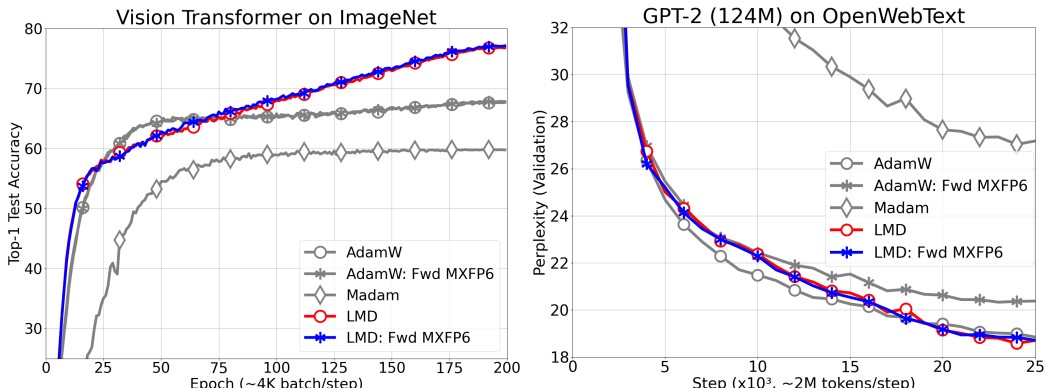

Figure 1: These two plots show that LMD trains Vision Transformer (ViT) and GPT-2 from scratch without performance loss, even with low-precision (MXFP6) forward passes. AdamW and Madam are unable to sufficiently train ViT. For GPT-2, AdamW shows perplexity similar to LMD, but performs worse under MXFP6. In GPT-2 training, LMD and Madam use a sequence length 4096 and a batch size 16; AdamW uses 1024 and 64, with equal tokens per step. All training used bfloat16 except the MXFP6 forward passes.

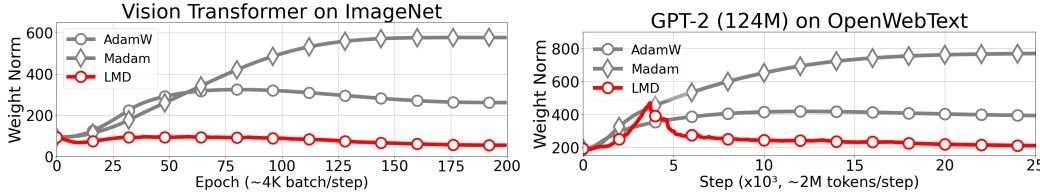

Figure 2: These two plots show the dynamics of the $\ell_2$ norm of weights during training of the models shown in Figure 1 when only using bfloat16. Madam takes a large norm, while LMD converges to a norm close to the initial value, clearly suppressing excessive weight increase.

Trinh et al., 2024). Notably, neither mechanism explicitly assumes log-normal weights. Zhong et al. (2022) studied log-normal weight learning but did not use multiplicative fluctuation properties.

In this paper, we propose the Log-Normal Multiplicative Dynamics (LMD) algorithm. LMD is derived by the Lie-Group Bayesian Learning Rule (Lie-Group BLR) with the multiplicative group of positive reals and log-normal weight noise distribution (Kiral et al., 2023). LMD explicitly assumes weights follow a log-normal distribution and uses MWU with both noise and regularization applied multiplicatively. We also incorporate the Lion (Chen et al., 2023) like gradient accumulation of the multiplicative gradients into the Lie-Group BLR, which stabilizes the dynamics. These combinations stabilize deep learning from scratch, see Figure 1. Our contributions are as follows:

1. We achieve the first successful training-from-scratch of ViT and GPT-2 using MWUs. Using low-precision emulation, we demonstrate that LMD can maintain performance under low-precision computation (Figure 1, Table 1, 2).

2. We show that LMD effectively suppresses the excessive weight growth that has been an issue for standard MWU methods (Figure 2). Through ablation studies, we verify that multiplicative weight decay strongly suppresses weight growth and delivers superior regularization compared to conventional additive weight decay (Figure 3).

3. We show that multiplicative noise injection is important for stabilizing the weights when using low-precision MX data format for forward passes (Figure 4). (Backward passes use bfloat16 (Kalamkar et al., 2019) rather than the MX data format.)

Table 1: ViT Performance comparisons; mean ± standard deviation (n = 3 independent runs). LMD uses sampled weight values during training and the expected value of the weight during testing.

| Dataset/ Architecture | Optimizer | Forward Precision | Train Accuracy (%) Top1 / Top5 | Test Accuracy (%) Top1 / Top5 | Weight Norm |
|---|---|---|---|---|---|
| ImageNet/ Vision Transformer | AdamW | bfloat16 | $93.67_{\pm 0.06}/97.83_{\pm 0.03}$ | $68.11_{\pm 0.38}/86.63_{\pm 0.21}$ | $260.7_{\pm 0.5}$ |
| | | MXFP6 | $93.41_{\pm 0.12}/97.73_{\pm 0.04}$ | $67.98_{\pm 0.27}/86.54_{\pm 0.15}$ | $261.5_{\pm 0.4}$ |
| | Madam | bfloat16 | $84.36_{\pm 0.03}/94.27_{\pm 0.02}$ | $60.14_{\pm 0.31}/81.45_{\pm 0.18}$ | $577.3_{\pm 0.9}$ |
| | | MXFP6 | $12.11_{\pm 0.17}/27.88_{\pm 0.30}$ | $13.47_{\pm 0.16}/30.59_{\pm 0.27}$ | $72.6_{\pm 0.4}$ |
| | LMD (Proposed) | bfloat16 | $79.84_{\pm 0.16}/93.53_{\pm 0.06}$ | $\mathbf{77.06_{\pm 0.08}/93.35_{\pm 0.04}}$ | $55.2_{\pm 0.1}$ |
| | | MXFP6 | $79.74_{\pm 0.34}/93.50_{\pm 0.14}$ | $\mathbf{77.15_{\pm 0.08}/93.42_{\pm 0.05}}$ | $55.5_{\pm 0.3}$ |

Table 2: GPT-2 Performance comparisons; mean ± standard deviation (n = 3 independent runs). LMD uses the expected value of the weight during validation evaluation.

| Dataset/ Architecture | Optimizer | Forward Precision | seq-len 1024 | | seq-len 4096 | |
|---|---|---|---|---|---|---|
| | | | Val. Loss | Weight Norm | Val. Loss | Weight Norm |
| OpenWebText/ GPT-2 | AdamW | bfloat16 | $2.937_{\pm 0.000}$ | $392.7_{\pm 0.4}$ | $4.791_{\pm 2.018}$ | $931.7_{\pm 70.5}$ |
| | | MXFP6 | $3.015_{\pm 0.000}$ | $424.9_{\pm 0.3}$ | $2.980_{\pm 0.002}$ | $426.1_{\pm 0.2}$ |
| | Madam | bfloat16 | $3.315_{\pm 0.001}$ | $762.4_{\pm 0.6}$ | $3.303_{\pm 0.003}$ | $765.1_{\pm 3.4}$ |
| | | MXFP6 | $4.620_{\pm 0.006}$ | $212.4_{\pm 1.3}$ | $4.575_{\pm 0.015}$ | $221.2_{\pm 1.5}$ |
| | LMD (Proposed) | bfloat16 | $2.961_{\pm 0.002}$ | $217.2_{\pm 4.5}$ | $\mathbf{2.925_{\pm 0.006}}$ | $212.9_{\pm 2.1}$ |
| | | MXFP6 | $2.971_{\pm 0.001}$ | $215.3_{\pm 2.9}$ | $\mathbf{2.927_{\pm 0.002}}$ | $216.9_{\pm 0.9}$ |

## 2 BACKGROUND AND RELATED WORK

### 2.1 MULTIPLICATIVE WEIGHT UPDATES

The low reliability of synaptic transmission is at the heart of neural computation (Seung, 2003). The brain not only overcomes such unreliability, but could even use it to drive learning (Kappel et al., 2015; Aitchison et al., 2021; Kasai et al., 2021). Synaptic transmission is driven by dynamic processes among the molecules that make up the synapse, and probabilistic approaches could be effective in understanding their robustness and plasticity (Choquet & Triller, 2013). Furthermore, spine sizes are thought to change multiplicatively (Loewenstein et al., 2011) and to store information logarithmically (Bartol et al., 2015), providing biological motivation for MWUs in ANNs (Bernstein et al., 2020; Pogodin et al., 2024; Cornford et al., 2024).

In general, MWUs attracted attention in various fields (Arora et al., 2012) such as machine learning (Littlestone, 1988; Kivinen & Warmuth, 1997), optimization (Freund & Schapire, 1997), game theory (Grigoriadis & Khachiyan, 1995). For a loss function $\ell(\boldsymbol{\theta})$, in contrast to gradient descent (GD), MWU modify the weights $\boldsymbol{\theta}$ via multiplication, for instance, consider the following versions:

$$\text{GD: } \boldsymbol{\theta} \leftarrow \boldsymbol{\theta} - \eta \nabla \ell(\boldsymbol{\theta}) \qquad \text{vs.} \qquad \text{MWU: } \boldsymbol{\theta} \leftarrow \boldsymbol{\theta} \odot \exp(-\eta \nabla \ell(\boldsymbol{\theta}) \odot \text{sign}(\boldsymbol{\theta})). \qquad (1)$$

Here, $\odot$ is an element-wise product, $\text{sign}(\boldsymbol{\theta})$ is a vector of $-1$, $0$ and $1$ depending on the sign of the entries, $\eta > 0$ is a learning rate, and exponentiation operations are performed element-wise. Bernstein et al. (2020) applied the above MWU to ANNs and claim that an advantage over GD is that the size of the update is proportional to the magnitude of the weight, which they claim has stabilizing properties. However, in their experiments, Madam tended to overfit, since multiplicative updates can increase the weights exponentially. They proposed weight clipping to stabilize and regularize training. The clipping threshold requires tuning, which motivates seeking other ways to stabilize training and avoid overfitting (Bernstein et al., 2020, Section 8).

### 2.2 MULTIPLICATIVE NOISE INJECTION

Biological synapses exhibit proportional changes in both activation-dependent plasticity and activation-independent spontaneous fluctuations, as discussed by Loewenstein et al. (2011). The

former corresponds to multiplicative update dynamics. The latter spontaneous multiplicative fluctuations can be modeled as a log-normally distributed noise injection. This is in tandem with experimental findings which show that the spine size observed in biological synapses follow a log-normal distribution (Loewenstein et al., 2011). There is no corresponding noise used in the Madam update (Bernstein et al., 2020), but noisy fluctuations may play an important role in suppressing excessive weight growth. This motivates the use of noise injection in ANN training.

Adding noise to the weights of ANNs during training also improves the generalization performance (An, 1996; Orvieto et al., 2022). Noise injection proportional to the magnitude of the weights has recently also shown impressive results for generalization (Bisla et al., 2022), as well as for improved robustness to corruptions (Trinh et al., 2024). However, these works use normal distributions which are unnatural in a multiplicative setting. Instead, sampling from the log-normal distribution,

$$q(\theta) = \text{LogN}(\theta \,|\, \mu, \sigma^2) := \frac{1}{\theta \sigma \sqrt{2\pi}} \exp\left(-\frac{(\log \theta - \mu)^2}{2\sigma^2}\right), \quad (2)$$

is more natural. The figure shows a normal distribution $N(\mu, \sigma^2)$ and a log-normal distribution $\text{LogN}(\mu, \sigma^2)$. Both use fixed $\sigma$ with two $\mu$ values. Here $m = \exp(\mu)$ is the median of the log-normal distribution. Sampling from a log-normal incorporates proportional noise injection, since if $\varepsilon \sim \text{LogN}(0, \sigma^2)$, then $m\varepsilon \sim \text{LogN}(\log m, \sigma^2)$. The standard deviation of $m\varepsilon$ is,

$$\text{std}[m\varepsilon] = \sqrt{\text{Var}[m\varepsilon]} = m e^{\frac{\sigma^2}{2}} \sqrt{\left(e^{\sigma^2} - 1\right)}, \quad (3)$$

which is proportional to its mean $\mathbb{E}[m\varepsilon] = m e^{\frac{\sigma^2}{2}}$ for fixed $\sigma$. We propose to use the log-normal distribution as an approximate posterior in variational learning, which we introduce next. We will see that this yields more natural multiplicative dynamics with inherent multiplicative noise injection.

## 2.3 VARIATIONAL LEARNING FOR NEURAL NETWORKS VIA LIE-GROUP UPDATE

Variational learning is formulated as an optimization problem over probability distributions $q(\boldsymbol{\theta})$,

$$\min_{q \in \mathcal{Q}} \mathbb{E}_{\boldsymbol{\theta} \sim q}[\ell(\boldsymbol{\theta})] + \tau \mathbb{D}_{\text{KL}}(q(\boldsymbol{\theta}) \,\|\, p_0(\boldsymbol{\theta})), \quad (4)$$

where $p_0(\boldsymbol{\theta}) \propto \exp(-R(\boldsymbol{\theta}))$ is a prior, $R(\boldsymbol{\theta})$ is a regularizer, $\tau > 0$ is a temperature parameter and $\ell(\boldsymbol{\theta}) = \sum_{i=1}^{N} \ell_i(\boldsymbol{\theta})/N$ denotes the empirical risk. For $\tau = 1$ and $\ell_i(\boldsymbol{\theta}) = -N \log p(\mathcal{D}_i \,|\, \boldsymbol{\theta})$, this problem targets the Bayesian posterior (Zellner, 1988), where $\mathcal{D}_i$ denotes the i-th training example. $\mathcal{Q}$ is the set of candidate distributions, and by minimizing Eq. 4 one searches for the candidate in $\mathcal{Q}$ closest to the exact Bayesian posterior, where closeness is measured by the Kullback-Leibler divergence. Variational learning for neural networks is implemented using stochastic gradient methods which leads to methods with noise injection (Graves, 2011; Blundell et al.; Khan & Rue, 2023).

In the past, multiplicative noise injection has been considered for Bayesian learning (Wen et al., 2018) but is typically used with additive GD updates. Recently, multiplicative updates have been derived from a variational viewpoint by using distributions parametrized via Lie-groups (Kiral et al., 2023). We use this framework to derive a new update because it naturally combine multiplicative updates with multiplicative noise injection. We start with Algorithm 2 in Kiral et al. (2023, App. A.4) which optimizes Eq. 4 using the following updates:

$$\boldsymbol{g} \leftarrow \boldsymbol{\theta} \odot (\nabla \ell(\boldsymbol{\theta}) + \tau \nabla R(\boldsymbol{\theta})) - \tau, \text{ where } \boldsymbol{\theta} \leftarrow \boldsymbol{m} \odot \boldsymbol{\varepsilon} \text{ and } \boldsymbol{\varepsilon} \sim \text{LogN}(\boldsymbol{\varepsilon} \,|\, 0, \sigma^2 \mathbf{I}),$$

$$\boldsymbol{\nu} \leftarrow \beta \boldsymbol{\nu} + (1 - \beta)\boldsymbol{g}, \quad (5)$$

$$\boldsymbol{m} \leftarrow \boldsymbol{m} \odot \exp\left(-\eta \boldsymbol{\nu} \odot \text{sign}(\boldsymbol{m})\right).$$

Here, exponentiation operations are element-wise, and $\beta \in [0, 1)$ a momentum parameter. From the viewpoint of Lie-groups, $\boldsymbol{g}$ and $\boldsymbol{\nu}$ are tangent vectors and the $\exp$-function is *the* natural mapping connecting the tangent space to the parameter space. Experimentally, they approximate $\boldsymbol{g} \approx \frac{1}{S} \sum_s \boldsymbol{\theta}^{(s)} \odot (\nabla \ell(\boldsymbol{\theta}^{(s)}) + \tau \nabla R(\boldsymbol{\theta}^{(s)})) - \tau$, $\boldsymbol{\theta}^{(s)}$ denotes the $s$-th Monte Carlo (MC) weight sampe from $q(\boldsymbol{\theta})$ and $S$ is the total number of MC samples. We refer to Kiral et al. (2023, Sec. 3.5) for details on the mathematical background and remark that the above method can also be viewed as gradient descent on the $\mu$-parameter of the log-normal distribution in Eq. 2.

We specialized the updates in Kiral et al. (2023, App. A.4) to the case of $\mathcal{Q}$ being the set of log-normal distributions from Eq. 2 with fixed $\sigma^2 \mathbf{I}$ and update $\boldsymbol{\mu}$ through $\boldsymbol{m} = \exp \boldsymbol{\mu}$, where $\boldsymbol{m} \in \mathbb{R}_{>0}^P$ is the median of the log-normal distribution and belongs to the multiplicative Lie-group that parametrizes the distribution. In contrast to the widely used MWU in Eq. 1, the updates in Eq. 5 (i) use noise injection, (ii) incorporate an explicit regularizer $R(\boldsymbol{\theta})$ and (iii) multiply $\nabla \ell(\boldsymbol{\theta})$ with the weight $\boldsymbol{\theta}$. In particular, we clarify the difference in the last feature. The classical MWUs setting obtains their gradients from considering additive perturbations $\ell(\boldsymbol{\theta} + t\epsilon)$, for which the derivative at $t = 0$ yields $\nabla \ell(\boldsymbol{\theta}) \odot \epsilon$. In contrast, Kiral et al. (2023) considers multiplicative perturbations consistent with the multiplicative geometry of the parameter space. These perturbations are of the form $\ell(\boldsymbol{\theta} \odot \exp(t\epsilon)) \approx \ell(\boldsymbol{\theta} \odot (1 + t\epsilon))$, so that the derivative at $t = 0$ yields $\nabla \ell(\boldsymbol{\theta}) \odot \boldsymbol{\theta} \odot \epsilon$. Therefore, the scaling of the gradient by the weight magnitude is a natural consequence of the type of perturbations they consider, from among which they choose the direction of fastest descent. Eq. 5 is the starting point for our proposed method, written in Algorithm 1.

A limitation of training ANNs with MWUs in Eqs. 1 and 5 is that the updates preserve each weight's sign. When signs are fixed, the expressive power of the network is limited (Bernstein et al., 2020, Section 8). To address this, we use the "EG± trick", representing a single weight as $\boldsymbol{\theta}_{\text{trick}} = \boldsymbol{\theta}_+ - \boldsymbol{\theta}_-$ with $\boldsymbol{\theta}_+, \boldsymbol{\theta}_- \geq 0$. This trick was also used by Kiral et al. (2023) and Kivinen & Warmuth (1997); Ghai et al. (2020). This form models the combined output of excitatory and inhibitory neurons, see (Kiral et al., 2023, Figure 1c). A biological interpretation of this learning rule only updating the magnitudes of weights respects Dale's law for ANNs (Amit et al., 1989).

## 2.4 STABLE LOW-PRECISION TRAINING VIA STOCHASTIC FLUCTUATION

Adopting low-precision data formats can substantially improve the throughput of matrix multiplications (GEMMs) (Courbariaux et al., 2015; Kalamkar et al., 2019; Micikevicius et al., 2022). Furthermore, attempts have been made to significantly improve power efficiency through co-design approaches of data formats and dedicated hardware units (Micikevicius et al., 2018; Wang et al., 2018; Peng et al., 2023; Wang et al., 2025). While quantization is expected to improve the efficiency of Transformer (Vaswani et al., 2017) pre-training, quantization based pre-training tends to suffer performance degradation (Chitsaz et al., 2024). To improve the efficiency of computationally intensive pre-training, a learning algorithm that is robust to low-precision training is necessary.

Multiplicative nature of log-normal weight distribution (discussed in Sec. 2.2) is compatible with numerical uncertainty in a way that is suitable for low-precision data formats. Rounding errors of floating-point and MX formats are roughly proportional to the magnitude of the values. Bernstein et al. (2020) also demonstrates the coherence of multiplicative weight updates with low-bit logarithmic data formats. Zhao et al. (2022) points out that for additive weight updates used in logarithmic formats (Miyashita et al., 2016), the impact of rounding errors increases with weight value, which leads to instability in learning.

For matrix multiplication, stochastic perturbations similar to log-normally distributed noise are empirically known to stabilize low-precision training. Stochastic rounding is a technique for inducing fluctuations in the rounding process to stabilize training (Gupta et al., 2015). This rounding randomly rounds a number to the nearest representable lower precision value, such that the expected value of the rounded result is equal to the original number (Croci et al., 2022). This has been shown to enable stable low-precision training in small networks (Gupta et al., 2015; Zhang et al., 2022). Even in LLMs, training-from-scratch using MX data format (Rouhani et al., 2023a) with stochastic rounding in the backward pass tends to retain performance (Tseng et al., 2025). Stochastic rounding has also been adopted for fine-tuning low-bit model ensembles (Nam & Lee, 2024).

In the quantization step following our multiplicative noise injection, values may occasionally be rounded to points farther from their nearest-neighbor. Considering that nearest-neighbor rounding in quantization does not always yield optimal results (Nagel et al., 2020), our work demonstrates that the stochasticity introduced by this noise injection may stabilize learning in low-precision formats, possibly via a mechanism analogous to stochastic rounding.

**Microscaling (MX) Data Formats** Microscaling (MX) data formats (Rouhani et al., 2023a) is one family of the proposed formats in Rouhani et al. (2023b) with the aim of achieving wide dynamic range with limited bit width. In the MX data formats a collection of $k_{\text{mx}}$ numbers is represented

---

**Algorithm 1** Proposed Log-Normal Multiplicative Dynamics (LMD) Optimizer

---

**Require:** Learning rate $\eta > 0$, temperature $\tau > 0$ , $\beta_1, \beta_2 \in [0, 1)$,

1: Initialize $\boldsymbol{m} = \left[\begin{smallmatrix} \boldsymbol{m}_+ \\ \boldsymbol{m}_- \end{smallmatrix}\right] \in \mathbb{R}^{2P}_{>0}$ by Sec. 3.2, $\boldsymbol{\nu} = \left[\begin{smallmatrix} \boldsymbol{\nu}_+ \\ \boldsymbol{\nu}_- \end{smallmatrix}\right] = \boldsymbol{0} \in \mathbb{R}^{2P}$ and $\mathbf{A}^\top = \left[\begin{smallmatrix} \mathbf{I} \\ -\mathbf{I} \end{smallmatrix}\right] \in \mathbb{R}^{2P \times P}$

2: **while** not converged **do**

3:     $\boldsymbol{\varepsilon} \sim \mathrm{LogN}(0, \sigma^2 \mathbf{I}_{2P})$                                    # sample log-normal noise

4:     $\boldsymbol{\theta} \leftarrow \boldsymbol{m} \odot \boldsymbol{\varepsilon}$ where $\odot$ is element-wise product            # multiplicative noise injection

5:     $\boldsymbol{g} \leftarrow \boldsymbol{\theta} \odot \mathbf{A}^\top \hat{\nabla} \ell(\boldsymbol{\theta}_{\mathrm{trick}})$ where $\boldsymbol{\theta}_{\mathrm{trick}} = \mathbf{A}\boldsymbol{\theta}$     # compute loss gradient using EG± trick

6:     $\boldsymbol{r} \leftarrow \tau(\boldsymbol{\theta} \odot \nabla R(\boldsymbol{\theta}) - 1)$                        # compute regularization gradient

7:     $\boldsymbol{\nu} \leftarrow \beta_2 \boldsymbol{\nu} + (1 - \beta_2)\boldsymbol{g}$ and $\boldsymbol{\nu}_{\mathrm{temp}} \leftarrow \beta_1 \boldsymbol{\nu} + (1 - \beta_1)\boldsymbol{g}$          # update momentum

8:     $\boldsymbol{m} \leftarrow \boldsymbol{m} \odot \exp\left(-\eta\left(\mathrm{sign}(\boldsymbol{\nu}_{\mathrm{temp}}) + \boldsymbol{r}\right)\right)$       # multiplicative update of median

9: **end while**

---

jointly, using very low precision private parts $p_{\mathrm{mx}}$ and and a shared integer exponent $s_{\mathrm{mx}}$. In the standard MX settings, $k_{\mathrm{mx}} = 32$, and the shared scale variable $s_{\mathrm{mx}}$ is of type INT8. For each private element $p_{\mathrm{mx}}$ we represent the numbers $2^{s_{\mathrm{mx}} - b_{\mathrm{mx}}} p_{\mathrm{mx}}$ where $b_{\mathrm{mx}}$ is called a bias. For example in the MXFP6 and MXFP4 data formats the bias term is 1 and the private elements $p_{\mathrm{mx}}$ either use the FP6 (E2M3) representation with 1-bit sign, 2-bit exponent and 3-bit mantissa, or use the FP4 (E2M1) representation with 1-bit sign, 2-bit exponent, and 1-bit mantissa. MX data formats have also been considered for deployment of LLMs because it enables improved throughput of matrix multiplications (Rouhani et al., 2023c; Verrilli, 2024; Tseng et al., 2025; Ramani et al., 2025). In addition, adopting the MX data format for deep learning models will contribute to the future energy-efficient hardware. For example, MXFP6-based computation system is estimated to provide approximately a $2\times$ advantage in hardware cost compared to FP8 (Rouhani et al., 2023b).

## 3 Log-Normal Multiplicative Dynamics for Deep Learning

We extend Eq. 5 to incorporate a design similar to synaptic multiplicative dynamics to enable stable low-precision training in large networks to overcome the small-scale limitation of Kiral et al. (2023). They only evaluated MWU assuming a Rayleigh distribution over the weights and therefore do not provide insight into actual implementations assuming a log-normal distribution.

Algorithm 1 shows the LMD Optimizer (the main differences from Eq. 5 are highlighted in red). As Loshchilov & Hutter (2019) show, incorporating gradient and weight penalties with momentum accumulation tends to destabilize learning and degrade performance. This instability arises because, when either the gradient or the penalty dominates, the accumulated moment neglects the other term (Bjorck et al., 2021). LMD addresses this issue by decoupling the gradient $\boldsymbol{g}$ and the regularizer $\boldsymbol{r}$ as in step 5 and 6 of Algorithm 1 as used in AdamW. We also use signed gradient momentum (Bernstein et al., 2018) and independent weight penalties, alongside two momentum coefficients, $\beta_1$ and $\beta_2$, as in Lion (Chen et al., 2023). The default exponential moving average (EMA) coefficient for momentum $\boldsymbol{\nu}$ is $\beta_2 = 0.99$ in step 7 of Algorithm 1. In each update, we interpolate between the current gradient $\boldsymbol{g}$ and the momentum $\boldsymbol{\nu}$ using $\beta_1 = 0.95$, and call that interpolation $\boldsymbol{\nu}_{\mathrm{temp}}$. We then apply a sign operation to yield the update rule in step 8 of Algorithm 1:

$$\boldsymbol{m} \leftarrow \boldsymbol{m} \odot \exp\left(-\eta\left(\mathrm{sign}(\boldsymbol{\nu}_{\mathrm{temp}}) + \boldsymbol{r}\right)\right). \tag{6}$$

We assume a separable component-wise regularizer $R(\boldsymbol{\theta}) = \sum_{i=1}^{2P} \tilde{R}(\theta_i)$ and set $\tau^{-1} := \tilde{R}'(1) - 1$ as such that the individual entries of $\boldsymbol{r}$ are $r_i = 1$ when the entries of $\boldsymbol{\theta}$ are $\theta_i = 1$. This corresponds to soft clipping for weights $\boldsymbol{\theta}$ greater than 1 by combination of signed moments, thereby aligning the magnitude of $\mathrm{sign}(\boldsymbol{\nu}_{\mathrm{temp}})$ and $r$, reducing the risk of learning instability. We only need to maintain two variables, $\boldsymbol{m}$ and $\boldsymbol{\nu}$. Then, $P$ weight parameters require additional $P$ variables. EG± trick doubles the weights, so LMD holds $4P$ parameters in total. By contrast, AdamW keeps $P$ weights plus $2P$ momentum terms for a total of $3P$, meaning LMD uses one additional $P$-dimensional vector compared to AdamW.

LMD also uses MC sampling and can efficiently perform multi-MC samples which effectively reduce the variance (Kingma et al., 2015), in multi-GPU environments. When using such multi-MC

sampling, $g$ and $r$ as in step 5 and 6 of Algorithm 1 are redefined as:

$$g \leftarrow \frac{\sum_{j,s} \boldsymbol{\theta}_j^{(s)} \odot \mathbf{A}^\top \hat{\nabla}\ell(\mathbf{A}\boldsymbol{\theta}_j^{(s)})}{J \cdot S}, \; r \leftarrow \frac{\sum_{j,s} \tau(\boldsymbol{\theta}_j^{(s)} \odot \nabla R(\boldsymbol{\theta}_j^{(s)}) - 1)}{J \cdot S}, \tag{7}$$

where $J$ is the number of devices, and $S$ is the number of MC samples per device. We use a different random sample $\boldsymbol{\theta}_j^{(s)}$ on each device $j$ and for each MC sample $s$. LMD is implemented as a drop-in replacement for Adam, similar to IVON (Shen et al., 2024). For details, see Appendix A.

## 3.1 MULTIPLICATIVE WEIGHT DECAY

In LMD we perform weight decay by assuming a log-normal distributed prior in Eq. 4. We choose a component-wise log-normal prior $\tilde{R}(\theta) = -\log p_0(\theta) = -\log(\mathrm{LogN}(\theta \mid \log m_r, \sigma^2))$, which leads to the following regularizer gradient $r$ used in Eq. 6, here written for a single component:

$$r = \tau(\theta\tilde{R}'(\theta) - 1) = \tau\Big[\theta \underbrace{\frac{1}{\theta}\Big(1 + \frac{\log\theta - \log m_r}{\sigma^2}\Big)}_{\tilde{R}'(\theta)} - 1\Big] = \tau\Big(\frac{\log\theta - \log m_r}{\sigma^2}\Big). \tag{8}$$

To understand the effect of the choice of the log-normal prior on the learning dynamics, we take the expectation of the regularizer gradient $r$ multiplied with the learning rate $\eta$. This leads to, $\mathbb{E}[-\eta\, r] = \mathbb{E}[-\alpha\log(\frac{\theta}{m_r})] = \log((\frac{m}{m_r})^{-\alpha})$ where $\alpha = \frac{\eta\tau}{\sigma^2} \geq 0$ and where we used that $\log\theta$ is distributed normally with mean $\log m$. Using these, the LMD update can be expressed as:

$$\boldsymbol{m} \leftarrow (\boldsymbol{m}^{1-\alpha}\, m_r^\alpha) \odot \exp\big(-\eta\,\mathrm{sign}(\boldsymbol{\nu}_{\mathrm{temp}})\big), \tag{9}$$

$$\Leftrightarrow \log\boldsymbol{m} \leftarrow (1-\alpha)\log\boldsymbol{m} + \alpha\log m_r - \eta\,\mathrm{sign}(\boldsymbol{\nu}_{\mathrm{temp}}). \tag{10}$$

As can be seen from Eq. 10, this simply corresponds to a weight decay in logarithmic space. Note that unlike usual weight decay, this penalty does not force weights to zero. Instead, it makes $m$ gravitate towards $m_r$. When used in conjunction with the plus-minus trick, then positive and negative weights move to $m_r$ and the expected value of $\boldsymbol{\theta}_+ - \boldsymbol{\theta}_-$ is zero. Larger values of $m_r$ lead to larger multiplicative noise injection but we still have zero-mean in expectation. As shown in Trinh et al. (2024), multiplicative weight noise is thought to mimic activation fluctuations like data augmentations:

$$z_j = \sum_i \big(m_{ji}(1 + \sigma\varepsilon_{ji})\big)x_i = \sum_i m_{ji}\big(\underbrace{(1 + \sigma\varepsilon_{ji})x_i}_{\text{perturbed activation}}\big), \tag{11}$$

where $x_i$ is the input (activation), $z_j$ is the $j$-th pre-activated neuron and $\varepsilon_{ji} \sim N(0,1)$ is the injected noise. Therefore, both weights that do not contribute to inference and remain near $m_r$ can be thought of as encouraging the emulation of activations rather than discarding them as in zero (pruned) weights.

## 3.2 INITIALIZATION

We also adjust the initialization to facilitate parameter exploration in the vicinity of $m_r$. Denote by $\theta_0$ the default initiailization of the neural network, and define the entires of $\boldsymbol{m}$ as:

$$m_{+,0} = \begin{cases} \theta_0\exp\big(-\frac{\sigma^2}{2}\big) + m_r, & \theta_0 > 0, \\ m_r, & \theta_0 \leq 0, \end{cases} \quad m_{-,0} = \begin{cases} m_r, & \theta_0 > 0, \\ -\theta_0\exp\big(-\frac{\sigma^2}{2}\big) + m_r, & \theta_0 \leq 0. \end{cases} \tag{12}$$

This ensures that the mean of the values of $\mathbf{A}\theta$ in line 5 of Algorithm 1 is $\theta_0$. That is, $\theta_0 = \mathbb{E}_{q(\theta_+|m_{+,0},\sigma^2)}[\theta_+] - \mathbb{E}_{q(\theta_-|m_{-,0},\sigma^2)}[\theta_-]$. Moreover, this allows us to reuse default initialization schemes which were devised for gradient-descent learning. For any parameters initialized to one, for instance, in scale parameters in normalization layers, we instead set $m_{+,0} = \exp(-\sigma^2/2)$, $m_{-,0} = 0$, $m_r = \exp(-\sigma^2/2)$, $\tau^{-1} = 2\tilde{R}'(2) - 1$. That is, the scale parameter is set so that the mean is 1 and the maximum is soft clipped to 2, which is heuristic but found to work well in practice and to stabilize the training.

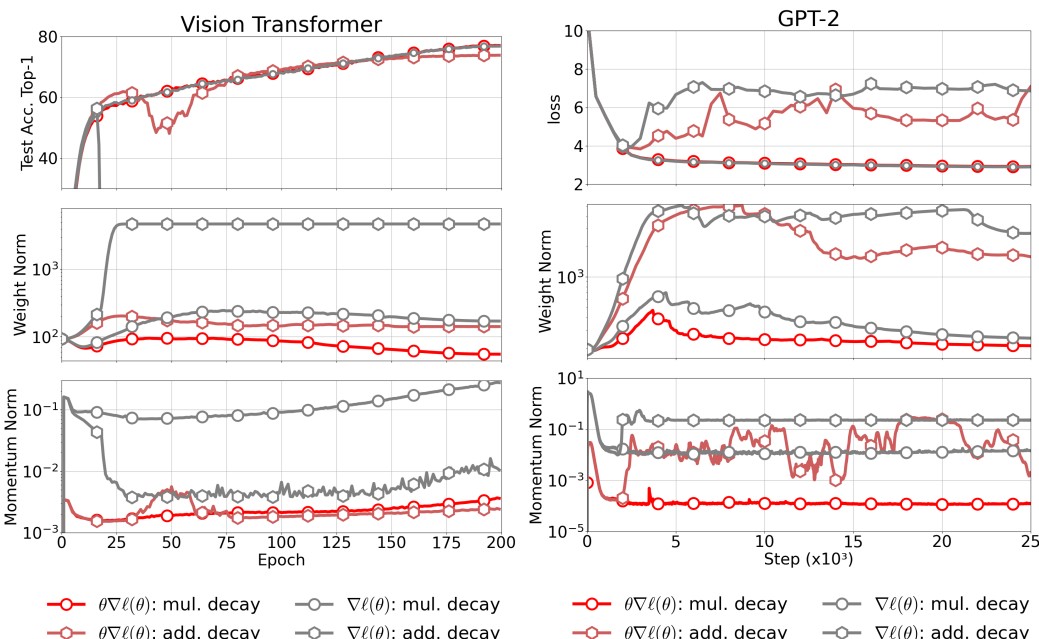

Figure 3: These plots show the effectiveness of multiplicative regularization in LMD. By scaling the gradient according to the weight magnitude and employing multiplicative weight decay in LMD, we observed consistently stable training dynamics in both ViT and GPT-2.

## 4 EXPERIMENTS

We first compare AdamW, Madam and LMD on training-from-scratch of ViT and GPT-2 in Sec. 4.1. We then validate the effectiveness of the multiplicative weight-decay regularizer derived in Eq. 9 (Sec. 4.2). Finally, we assess the role of multiplicative noise injection in enabling low-precision training (Sec. 4.3). All experiments using the MX Format were performed using software emulation. See Appendix B for detailed experimental settings. Additionally, in the Appendix C, we present a comparison of the original algorithm by Kiral et al. (2023) with LMD, as well as a comparison of Madam's well-performing small transformer setting with LMD. In both cases, LMD achieves superior performance.

### 4.1 COMPARISON WITH EXISTING OPTIMIZERS

We compared LMD against AdamW and Madam (Figure 1, 2). The detailed results are shown in Table 1, 2. In ViT training, LMD was shown to overwhelmingly outperform both AdamW and Madam (Table 1). In GPT-2 training, when the maximum sequence length (corresponding to "seq-len" in the Table 2) was 4096, LMD achieved higher performance than any of the AdamW variants. In MXFP6 forward-pass training, AdamW tended to perform poorly or produce inconsistent results, whereas LMD did not. This demonstrates that LMD can leverage low-precision forward computations yet still train both stably and with high accuracy. Madam performed worse in all settings, and as shown in Figure 2 its weight $\ell_2$ norm tended to grow relative to the other optimizers, indicating unstable training. LMD, in contrast, nearly converged to the initial weight $\ell_2$ norm, suggesting that it seeks optimal solutions in regions of parameter space where the weight norm remains small. The weight norms for LMD were computed by using the mean value of $\boldsymbol{\theta}_+ - \boldsymbol{\theta}_-$.

### 4.2 EFFECTIVENESS OF MULTIPLICATIVE REGULARIZATION IN LMD

We perform an ablation study to understand why LMD trains effectively. The regularization in LMD stems not only from the multiplicative weight update in Eq. 1 but also from scaling the gradient by the weight, as shown in Eq. 5. We show that these two features suppress the excessive weight

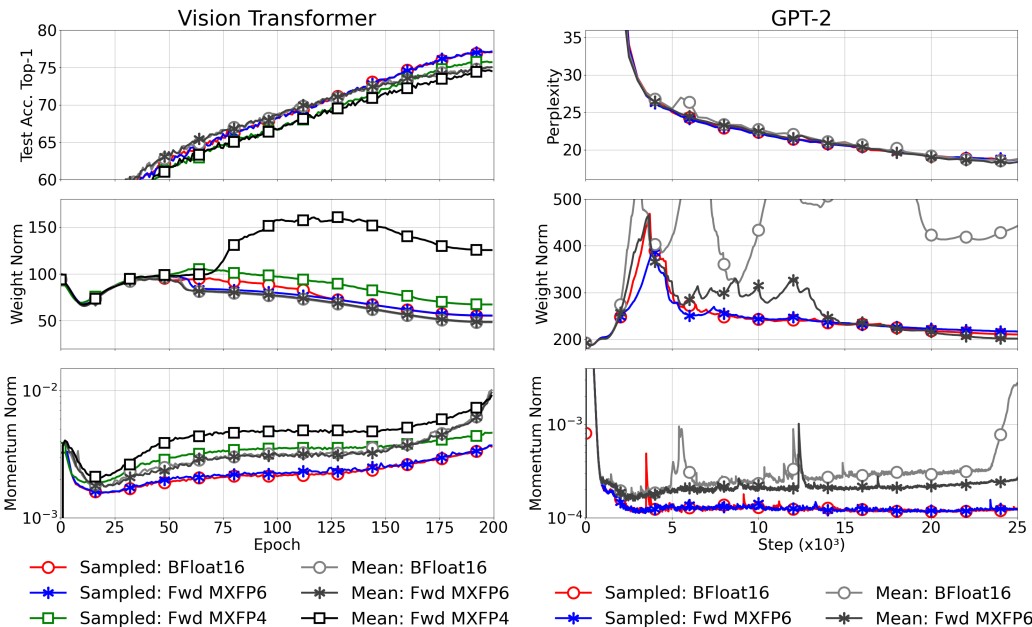

Figure 4: This panel shows the effectiveness of multiplicative noise injection for low-precision training. By sampling weights, it is possible to perform weight regularization and gradient stabilization.

growth characteristic of multiplicative updates. Figure 3 depicts a total of four cases: multiplicative weight decay or additive weight decay (i.e. gaussian prior), each combined with either gradient scaling by weights or no scaling. Empirically, since the momentum $\nu$ does not change the trend based on the sign, we plot the $\ell_2$ norm of the momentum for positive weights. The momentum norm captures the dynamics of the gradient magnitudes through accumulation of past gradients. Additive weight decay was evaluated with $r = (\theta - m_r)/(1 - m_r)$. This means that when $\theta$ is one, $r$ also one. Overall, multiplicative weight decay contributes more strongly to stabilizing LMD training compared to additive decay. Under additive decay, the weight $\ell_2$ norm tends to increase, indicating weak regularization, and this effect is especially pronounced in GPT-2. Under multiplicative decay, when the gradient is scaled by the weight, fluctuations in the weight $\ell_2$ norm are more gradual than in the unscaled case , clearly demonstrating effective regularization. The momentum norm is also visibly smaller in magnitude, indicating suppressed gradient variance.

## 4.3 EFFECTIVENESS OF MULTIPLICATIVE NOISE FOR LOW-PRECISION TRAINING

We investigated the importance of multiplicative noise injection in low-precision training to use MX data format emulation. Multiplicative weight noise suppresses the growth of the the momentum norm and yields consistent learning dynamics regardless of whether a MXFP6 forward pass is used. Figure 4 plots four cases in GPT-2: training with bfloat16 forward passes and MXFP6 forward passes, each combined with either sampled training or mean training. In the case of ViT, six patterns are shown by adding MXFP4 results. In the ViT experiments, sampled training clearly achieves higher accuracy than using only the mean training. This is because in the final phase of training (when cosine annealing has a strong effect), sampling better suppresses the growth of the $\ell_2$ norm of the momentum, which avoid overfitting. Looking at the MXFP4 results, the mean training (black) shows a noticeably larger $\ell_2$ norm of weights compared to the sampled training (green), indicating that sampling is an important regularizer in low-precision training. In the GPT-2 experiments, sampled training showed almost identical dynamics regardless of precision, demonstrating that LMD's multiplicative noise injection enables stable low-precision training. Conversely, mean training by MXFP6 showed a clear increase in $\ell_2$ norm of weights, indicating insufficient regularization. Nevertheless, the final performance of the mean training using MXFP6 was close to that of sampled training. This suggests that the numerical error in MXFP6 may have played a role in regularization, but the dynamics are inconsistent with the bfloat16 result.

## 5 DISCUSSION

This work is biologically motivated and asks whether multiplicative weight is feasible with low-precision deep learning. To evaluate optimizer stability, we focus on training-from-scratch at ViT and GPT-2 scale. Although we demonstrated stable training when using MXFP6 for forward passes, we have not evaluated low-precision backward passes, and because all experiments were conducted via emulation, we did not measure any real-world speedups by hardware that natively supports the MX data formats same as Rouhani et al. (2023c). In our preliminary experiments of LMD, low-precision backward passes did not perform well. One possible reason is that LMD applies (multiplicative) stochastic perturbations only to the weights, leaving the gradients unchanged. In contrast, low-precision backward passes can be trained effectively with stochastic rounding (SR) (Tseng et al., 2025), so combining LMD and SR is a promising direction. Low-precision forward passes using SR tend to diverge when used for training (NVIDIA et al., 2025, Figure 10), suggesting that LMD may play a complementary role in this setting. The EG± trick doubles the number of weight parameters that need to be learned, which may increase memory usage. However, in large transformers such as GPT, the memory usage bottleneck lies not in the weight parameters themselves but in the activations (Chitsaz et al., 2024; Chen et al., 2025). While the EG± trick doubles the number of weights, the number of weights used in matrix multiplication remains unchanged, so it does not affect the memory usage of the activations and computational costs of GEMMs. We have not investigated LMD for fine-tuning and other downstream tasks. We believe that our promising results LMD will motivate further research in these areas.

**Conclusions** We propose LMD, a multiplicative weight update (MWU) algorithm that incorporates the log-normal multiplicative dynamics observed in biological synapses. The multiplicative regularization introduced by LMD suppresses the excessive weight growth that was a challenge in MWU, making it possible to train large-scale neural networks such as ViT and GPT-2. Notably, even when using the low-precision MXFP6 data format for the forward pass, no performance degradation was observed. These results suggest that LMD has the potential to serve as an effective optimizer when utilized with energy-efficient hardware architectures designed for low-precision computations.

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

## A    SETTING DETAILS FOR THE LMD OPTIMIZER

**Pytorch Implementation**    LMD is easy to use because it is implemented as a pytorch optimizer, like AdamW and Madam, and its implementation is very similar to IVON (Shen et al., 2024)[1], an optimizer for Bayesian deep learning. The simplest way to use it is shown in the Listing 1.

---

**Listing 1** The simplest implementation of LMD in pytorch.

```
1   import torch
2  +from lmd.lmd import LMD
3
4   train_loader = torch.utils.data.DataLoader(train_dataset)
5   test_loader  = torch.utils.data.DataLoader(test_dataset)
6   model        = MLP()
7
8  -optimizer = torch.optim.AdamW(model.parameters())
9  +optimizer = LMD(model)
10
11  for X, y in train_loader:
12
13 +  for _ in range(train_samples):
14 +    with optimizer.sampled_params():
15       optimizer.zero_grad()
16       logit = model(X)
17       loss  = torch.nn.CrossEntropyLoss(logit, y)
18       loss.backward()
19
20     optimizer.step()
```

---

## B    EXPERIMENTAL DETAILS

**Microscaling (MX) Data Format Settings**    In this study, both MXFP6 and MXFP4 formats are used as default settings. To enable training of large networks on existing GPUs, we use a custom CUDA library to emulate the MX data format (Project, 2023; Rouhani et al., 2023c). Under this emulation, low-precision matrix multiplications are performed in MXFP6 or MXFP4. In our experiments, vector operations such as activations and all backward-pass computations are carried out in bfloat16. We note that the internal state variables of the optimizer still remains in FP32.

**MX Data Format Emulation**    We applied the MX PyTorch Emulation Library[2] for forward-only, low-precision experiments. In this emulation (see Figure 6), activations and weights are first converted to the specified MX data format and then dequantized to bfloat16. Matrix multiplication is performed on these dequantized values; the result is then cast to bfloat16, after which activation and normalization layers are applied. Backward propagation is carried out in bfloat16.

**Madam Settings**    As discussed by Bernstein et al. (2020), the performance of Madam varies significantly depending on the threshold of the weight clipping. Therefore, in this paper, we clip the magnitude of Madam's weights so that they do not exceed 1, consistent with the soft clipping applied to the weights of LMD. This is because, in Madam's original clipping settings, the initial $\ell_2$ norm of the weights drops dramatically in the first update regardless of the gradient, making comparisons of training dynamics with LMD or AdamW impossible.

**ViT Experimental Settings**    We employed a Vision Transformer (ViT) with an embedding dimension of $384$, 6 attention heads, and 12, transformer layers. Input images were resized to $224 \times 224$ pixels and split into $16 \times 16$ patches. Instead of using the [CLS] token embedding, we applied global average pooling over patch embeddings for classification. Positional embeddings were treated as

---

[1]https://github.com/team-approx-bayes/ivon
[2]https://github.com/microsoft/microxcaling/tree/v1.1.0

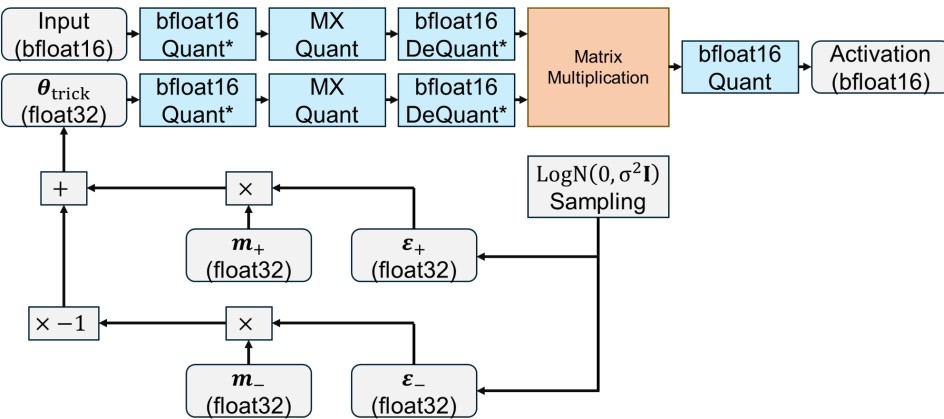

Figure 6: Overview of MX Pytorch Emulation for forward-pass with noise injection. Quantization (Quant) and Dequantization (DeQuant) of bfloat16 and quantization of the specified MX format are performed. The weights $m_+$ and $m_-$, as well as the sampled noises $\varepsilon_+$ and $\varepsilon_-$, were represented using float32. If you do not use emulation, the steps marked with * will be skipped.

learnable parameters. ViT adopts the implementation of PyTorch Image Models (Wightman, 2019). All ViT experiments were run on the ImageNet dataset (Deng et al., 2009) for 200 epochs with a batch size of 4,096. We applied standard data augmentation, random resized crop to $224 \times 224$ and random horizontal flip. Training was performed on eight NVIDIA H100 GPUs (96 GB each), requiring about 6 to 9 hours; when using the MX format emulator extended this to roughly half a day (Table 3). we distributed one GPU per node across eight nodes. In LMD experiments, each GPU samples weights from a different random seed, resulting in MC sampling beging $J = 8$ and $S = 1$. The learning rate follows a cosine-annealing schedule down to zero, with a linear warmup over the first 10,000 steps. For LMD, we set the learning rates to $\eta = 0.005, \sigma = 0.125$, and $m_r = 0.01 \times \exp\left(\frac{\sigma^2}{2}\right)^3$. LMD did not employ gradient-norm clipping. For AdamW, we used a learning rate of $0.001$, with $\beta_1 = 0.9$, $\beta_2 = 0.999$, and a weight decay of $0.1$. For Madam, the learning rate was $0.01$, and parameter values were clipped to a maximum magnitude of $1$. Both AdamW and Madam employed gradient-norm clipping at $1$.

Table 3: ViT training time comparisons; mean ± standard deviation (n = 3 independent runs). This training time includes the time required to acquire data, such as calculating the weight and momentum norm.

| Dataset/ Architecture | Optimizer | Forward Precision | Time (min) | Ratio |
|---|---|---|---|---|
| | AdamW | bfloat16 | $433.1_{\pm 36.8}$ | 1.0 |
| | | MXFP6 | $583.7_{\pm 2.1}$ | 1.35 |
| ImageNet/ Vision Transformer | Madam | bfloat16 | $426.4_{\pm 59.1}$ | 0.98 |
| | | MXFP6 | $607.5_{\pm 3.8}$ | 1.40 |
| | LMD (Proposed) | bfloat16 | $480.3_{\pm 69.8}$ | 1.11 |
| | | MXFP6 | $654.2_{\pm 27.6}$ | 1.51 |

**GPT-2 Experimental Settings** We trained the GPT-2 model Radford et al. (2019) on the Open-WebText dataset Gokaslan & Cohen (2019) for a total of 25,000 steps, processing approximately $2.1 \times 10^6$ tokens per step (total $\approx 52$ Billion tokens) using 32 gradient accumulations. We used the nanoGPT framework (Karpathy, 2022) to train the model. Training was performed on eight NVIDIA H100 GPUs (96 GB each), requiring about 5 to 7 hours; when using the MX format emulator extended this to roughly half a day but employed sixteen H100 GPUs (Table 4). We assigned one GPU to each node and determined the number of nodes based on the number of GPUs to be

---
[3] $m_r = 0.01$ is expected to give the same results.

used. In the LMD experiments, on each GPU, we sampled weights from a different random seed, and the number of MC samples per weight update was equal to the number of gradient accumulation steps with 32 times, i.e., $J = 8$ and $S = 4$ were used. The learning rate decays to one-tenth of its initial value via cosine annealing, following a linear warmup over the first 2,000 steps. We trained all GPT-2 models without bias parameters with 124 M parameters in total. For LMD, we set the initial learning rate to $\eta = 0.005, \sigma = 0.125$, and $m_r = 0.01 \times \exp\left(\frac{\sigma^2}{2}\right)$ (same as ViT settings). LMD employed gradient-norm clipping at 10. For AdamW, we used a learning rates of 0.0006, $\beta_1 = 0.9, \beta_2 = 0.95$, and a weight decay of 0.1. For Madam, the learning rate was $1 \times 10^{-2}$ with a weight decay of 0.1.

Table 4: GPT-2 training time comparisons; mean ± standard deviation (n = 3 independent runs). This training time includes the time required to acquire data, such as calculating the weight and momentum norm. [†] The MXFP6 experiments used 16 GPUs not 8 GPUs, so the times and ratios cannot be simply compared.

| Dataset/ Architecture | Optimizer | Forward Precision | seq-len 1024 | | seq-len 4096 | |
|---|---|---|---|---|---|---|
| | | | Time (min) | Ratio | Time (min) | Ratio |
| OpenWebText/ GPT-2 | AdamW | bfloat16 | $306.3_{\pm 3.6}$ | 1.0 | $385.3_{\pm 0.5}$ | 1.0 |
| | | MXFP6[†] | $620.9_{\pm 35.2}$ | 2.03 | $662.1_{\pm 34.7}$ | 1.72 |
| | Madam | bfloat16 | $302.4_{\pm 0.4}$ | 0.99 | $383.2_{\pm 2.0}$ | 1.00 |
| | | MXFP6[†] | $602.9_{\pm 0.7}$ | 1.97 | $643.0_{\pm 0.4}$ | 1.67 |
| | LMD (Proposed) | bfloat16 | $367.2_{\pm 7.3}$ | 1.20 | $443.1_{\pm 0.8}$ | 1.15 |
| | | MXFP6[†] | $649.2_{\pm 1.2}$ | 2.12 | $688.4_{\pm 0.7}$ | 1.79 |

## C  EXPERIMENTS FOR SMALL NETWORKS

### C.1  COMPARISON OF LMD AND KIRAL ET AL. (2023) IN FULLY CONNECTED NETWORKS

We evaluated the original Lie-Group Bayesian Learning Rule (LGBLR) algorithm of Kiral et al. (2023) using their publicly available JAX implementation[4], while we implemented LMD and AdamW in PyTorch under the same experimental settings. We trained a fully connected network with 5 hidden layers (1024, 512, 256, 256, 256 neurons) and tanh nonlinearity. We trained on MNIST for 25 epochs with a batch size of 50. Kiral et al. (2023) did not conduct experiments with a log-normal distribution; instead, their multiplicative weight updates used a Rayleigh distribution. In their setup, the learning rate was set to 50 and the number of Monte Carlo (MC) samples to 32. To match the conditions of LMD, we reduced the number of MC samples to 1 and changed the learning rate over 50, 20, and 10. For AdamW, we used a learning rate of $1 \times 0.001$ with $\beta_1 = 0.9$ and $\beta_2 = 0.999$. For LMD, we used a learning rate of 0.005, $\beta_1 = 0.95$, $\beta_2 = 0.999$, and $\sigma = 0.125$.

Table 5 shows that, under the original proposal of Kiral et al. (2023), training is difficult even for small networks. In their experiments, they were only able to successfully train MLPs by using the large number of MC samples to stabilize learning. When the MC size is reduced to 1, as in LMD, the method either diverges or exhibits a clear degradation in performance.

### C.2  COMPARISON OF LMD AND MADAM IN SMALL TRANSFORMERS

We compared LMD against Madam under experimental settings in which Madam can train stably. We trained a Transformer model with two hidden layers on the WikiText-2 dataset, using their publicly available implementation[5]. For Madam, we used a learning rate of 0.01. In the GPT-2 experiments, the weight was clipped to 1 to match soft-clipping of LMD; in this experiment,

---

[4]https://github.com/team-approx-bayes/liegroups
[5]https://github.com/jxbz/madam

Table 5: MLP performance comparisons; mean ± standard deviation (n = 3 independent runs).

| Dataset/ Architecture | Optimizer | Learning Rate | MC Sample | Test Accuracy (%) |
|---|---|---|---|---|
| MNIST/ Multi- Layer Perceptron | Original | 50 | 32 | $98.23_{\pm 0.03}$ |
| | Lie-Group | 50 | 1 | $NaN$ |
| | BLR | 20 | 1 | $85.63_{\pm 2.99}$ |
| | (Kiral et al., 2023) | 10 | 1 | $62.52_{\pm 1.72}$ |
| | AdamW | $1e-3$ | – | $98.25_{\pm 0.03}$ |
| | LMD (Proposed) | $5e-3$ | 1 | $\mathbf{98.60_{\pm 0.02}}$ |

however, we set the weight clipping parameter of Madam (called `scale`) to 2.0 in order to prioritize the performance of Madam. For AdamW, we used a learning rate of $0.0001$ with $\beta_1 = 0.9$ and $\beta_2 = 0.999$. For LMD, we used a learning rate of $0.005$, $\beta_1 = 0.95$, $\beta_2 = 0.99$, and $\sigma = 0.25$.

Table 6 shows that Madam achieves better performance than AdamW, while LMD further outperforms Madam in bfloat16 settings. Furthermore, similar to the ViT and GPT-2 experiments, LMD does not exhibit performance degradation due to low-precision forward passes (MXFP6).

Table 6: Small transformer performance comparisons; mean ± standard deviation (n = 3 independent runs). LMD uses the expected value of the weight during testing.

| Dataset/ Architecture | Optimizer | Forward Precision | Test PPL |
|---|---|---|---|
| Wikitext-2/ Transformer | AdamW | bfloat16 | $170.5_{\pm 0.3}$ |
| | | MXFP6 | $264.3_{\pm 81.0}$ |
| | Madam | bfloat16 | $165.9_{\pm 0.2}$ |
| | | MXFP6 | $387.2_{\pm 3.1}$ |
| | LMD (Proposed) | bfloat16 | $\mathbf{156.1_{\pm 0.5}}$ |
| | | MXFP6 | $\mathbf{155.9_{\pm 0.6}}$ |

## D  THE USE OF LARGE LANGUAGE MODELS (LLMS)

This paper utilized LLMs for the following purposes: as a machine translation tool for writing the paper; and as an aid for basic software implementation.

