# OpenReview forum: "Log-Normal Multiplicative Dynamics for Stable Low-Precision Deep Learning"
_ICLR.cc/2026/Conference — Submitted to ICLR 2026_

### Official Review · Reviewer_vFuX · 2025-10-31

**Soundness:** 2
**Presentation:** 2
**Contribution:** 2
**Rating:** 2
**Confidence:** 3

**Summary:**

This paper introduces LMD, a multiplicative weight update algorithm that draws on log-normal dynamics observed in biological synaptic spines. Experiments on ImageNet and OpenWebText show that LMD maintains stability and delivers robust performance even with low-precision forward passes. It also exhibits strong results when applied to GPT-2 and ViT models.

**Strengths:**

- This work achieves the first successful training of large networks using multiplicative weight updates (MWU). The proposed LMD algorithm even outperforms AdamW on ImageNet, clearly validating the effectiveness of the method.

**Weaknesses:**

- While this study builds primarily on the Lie-group Bayesian learning framework (Kiral et al., 2023), it fails to clearly delineate the key distinctions between LMD and the prior method, nor does it offer a direct performance comparison. The absence of such analysis undermines the apparent novelty and impact of the proposed approach.

- There is no theoretical support for why LMD achieves higher accuracy or performs well in low-precision scenarios.

- Low-precision training is not a particularly important issue. Fig. 1 also does not show that LMD has a substantial improvement over the common AdamW.

**Questions:**

- On the theoretical side, a discussion of why LMD is expected to work better than previous algorithms would be welcome.

-  LMD uses Monte Carlo (MC) sampling. Can you explain the impact of S on computational cost? I don’t quite understand what problem the introduction of MC solves.

- Why it is effective in large networks, or if it works in both large and small networks?

**Details Of Ethics Concerns:**

NO ethics concerns

---

> ### Author Response · Authors · 2025-11-25
>
> We thank the reviewers for their careful evaluation of our work and for pointing out that (i) LMD constitutes the first successful training of large networks using MWU, and (ii) in ImageNet training on ViT, LMD clearly achieves performance surpassing that of AdamW.
>
> We have now addressed the lack of comparison with the original method by Kiral et al., including a direct performance comparison. Taking these results into account, we would be grateful if you could reconsider your score.
>
> We have responded to the concerns raised below.
>
> > Comment 1: “While this study builds primarily on the Lie-group Bayesian learning framework (Kiral et al., 2023), it fails to clearly delineate the key distinctions between LMD and the prior method, nor does it offer a direct performance comparison.”
>
> Thank you for pointing out that the differences from the Lie-group Bayesian learning framework [Kiral et al., 2023] were not sufficiently clear. Kiral et al. do not experiment with log-normal distributions (they use Rayleigh instead), so a direct comparison in that setting is not possible. We clarify this in L299–300. We also highlight in red in Algorithm 1 the main changes relative to Kiral et al.
> We agree it is important to compare against their experiments, so we reproduced their Rayleigh setup and added results in Appendix C-1. Their method is difficult to use in practice: even MNIST training fails unless many Monte Carlo (MC) samples are used, and performance degrades clearly when using only one MC sample as in LMD. Please see our response to Q2-1 for details on MC samples and computational cost.
>
> > Comment 2: “There is no theoretical support for why LMD achieves higher accuracy or performs well in low-precision scenarios.”
>
> In this paper, we do not aim to provide a theoretical proof. However, we discuss why LMD enable to train networks effectively in Sec. 2.4, and we further argue in lines L353–363 that multiplicative noise injection can be interpreted as introducing perturbations on the activations like data augmentations [Trinh et al., 2024].
>
> > Comment 3-1: “Low-precision training is not a particularly important issue.”
>
> The research area of low-precision training is actively discussed as an important issue in the ICLR community. For example, the paper that investigated changing pre-training from FP32 to FP16 was accepted at ICLR 2018 and has received nearly 2,600 citations to date[Micikevicius et al., 2018]. Moreover, a recent study on scaling laws for low-precision pre-training and post-training was accepted as an Oral at ICLR 2025 [α]. Given this trend, we believe it is reasonable to regard low-precision training as a research direction that continues to attract attention at ICLR.
>
> [α] Tanishq Kumar, et al.,  Scaling Laws for Precision. In International Conference on Learning Representations, 2025.
>
> > Comment 3-2: “Fig. 1 also does not show that LMD has a substantial improvement over the common AdamW.”
>
> Figure 1 clearly shows that LMD outperforms AdamW on ViT. The GPT-2 results show that, unlike AdamW, LMD does not experience performance degradation when using low precision forward passes. This trend is further supported by additional experiments using smaller Transformer models, as shown in Table 6 in Appendix C.
>
> > Q1: “On the theoretical side, a discussion of why LMD is expected to work better than previous algorithms”
>
> As reviewer JeNJ pointed out (see comment 2 and our response), LMD employs a distinctly different update rule from classical MWU, and we agree that its theoretical analysis is an interesting direction. However, it is not the focus of this paper. To be convincing to the broader community, it is essential to demonstrate its applicability not only to MNIST but also to modern architectures such as ViT and GPT-2. We believe that our paper contributes precisely to this point.
>
> > Q2-1: “LMD uses Monte Carlo (MC) sampling. Can you explain the impact of S on computational cost?”
>
> When using MC sampling, random numbers must be generated for each weight, but the main computational cost comes from needing S forward and backward passes per update, not from sampling itself. Kiral et al. used S=32, incurring large overhead, whereas LMD achieves stable training with S=1 (see Appendix C-1).
>
> > Q2-2: “ I don’t quite understand what problem the introduction of MC solves.”
>
> Thank you for the feedback on improving the clarity of the paper. MC sampling is not specific to LMD. It is a standard technique in variational learning for neural networks. We have therefore added an explanation of MC sampling in Sec. 2.3 (L212–L214).
>
> Regarding its practical role, we experimentally tested in Figure 4 the hypothesis that MC sampling helps suppress the growth of the weight norm (L166-167).
>
>
> > Q3: “Why it is effective in large networks, or if it works in both large and small networks?”
>
> We have added results for small networks in Appendix C. LMD can be effectively trained even for small networks.

---

### Official Review · Reviewer_YL91 · 2025-10-31

**Soundness:** 2
**Presentation:** 2
**Contribution:** 1
**Rating:** 4
**Confidence:** 3

**Summary:**

This paper proposes the Log-normal Multiplicative Dynamics (LMD) algorithm for stable learning under low-precision computation (MXFP6).

**Strengths:**

1）This paper proposes the Log-normal Multiplicative Dynamics (LMD) algorithm for stable learning under low-precision computation and validated on vision and language models.

**Weaknesses:**

1)The paper lacks an evaluation of the actual training speed and memory consumption, likely because the proposed parameter optimization method is relatively complex and computationally intensive.

2)The ablation study is insufficient, as the main text does not include a quantitative comparison with the standard MWU method.

**Questions:**

1) The paper uses the low-precision MX data format for forward propagation and BF16 for gradient backpropagation. What would happen if the gradient backpropagation during training also used the MX data format?

2) How does the performance of this paper's model compare to that of a model trained with BF16 and deployed on low-precision hardware?

---

> ### Author Response · Authors · 2025-11-25
>
> We thank the reviewers for their review and for pointing out that  (i) LMD enables stable learning with low-precision forward computation.
>
> In addition to the stability of low-precision forward training, a key contribution of this paper is that LMD overcomes the scalability limitations of MWU. Taking these points into account, we would be grateful if you could kindly reconsider their score.
>
> We have responded to the concerns raised below.
>
> > Comment 1“The paper lacks an evaluation of the actual training speed and memory consumption, likely because the proposed parameter optimization method is relatively complex and computationally intensive.”
>
> We focus on evaluating the stability of the LMD optimizer rather than tuning it for speed, so we did not provide detailed measurements of training time or memory usage. Our implementation also includes extra overhead, such as momentum norm computation and MX data format emulation.
>
> However, we agree it is useful to report the computational cost needed to reproduce our experiments, and we have added this information to Tables 3 and 4 in Appendix B. In the bfloat16 setting, LMD is about 11–20% slower than AdamW in time, mainly due to the overhead of weight sampling. Although LMD is slightly slower than AdamW, it clearly outperforms ViT in Table 1, and GPT-2 training is expected to be faster by using low-precision forward passes.
>
> We also noticed that emulation was mentioned only in the Discussion and Appendix section. Thank you for pointing this out. We have now added this point at L409.
>
> > Comment 2 “The ablation study is insufficient, as the main text does not include a quantitative comparison with the standard MWU method.”
>
> Among existing multiplicative methods, Madam is the only one that has successfully trained Transformer architectures from scratch, so it is reasonable to use it as the standard MWU baseline. Madam is sufficient as an ablation to study MWU scalability, and note that even the Lie-Group BLR paper did not attempt Transformer training.
>
> Although Madam cannot train large Transformers such as ViT or GPT-2, it works relatively well on smaller models. We therefore added experiments comparing LMD and Madam in this favorable setting (Appendix C-2). LMD outperforms both Madam and AdamW, and this shows no performance degradation in LMD when using MXFP6 forward passes even on small Transformers.
>
> > Q1 “What would happen if the gradient backpropagation during training also used the MX data format?”
>
> Our preliminary experiments confirmed that using the MX data format for backpropagation does not work well for ViT and GPT-2. This is because LMD essentially injects noise only into the weights and does not apply stochastic perturbation to the gradients in the backward pass. When using the MX data format in the backward pass, stochastic rounding is considered an effective approach [Tseng et al., 2025]. Therefore, combining LMD with these techniques may enable low-precision learning in both the forward and backward passes. We believe this is an interesting direction for future research which we added in L497 . The low-precision forward passes by the SR approach tends to diverge [α, Figure10].
>
> [α] NVIDIA et al., Pretraining Large Language Models with NVFP4, arXiv preprint arXiv:2509.25149, 2025
>
>
> > Q2 “How does the performance of this paper's model compare to that of a model trained with BF16 and deployed on low-precision hardware?”
>
> For direct-cast inference, where a trained model is converted to the MX data format without tuning, MXINT8 is generally preferred, and MXFP6 causes slight performance degradation[Rouhani et al., 2023c]. On the other hand, if the model is trained using LMD with low-precision forwards, there is no need to consider the impact of direct-cast inference.

---

### Official Review · Reviewer_JeNJ · 2025-10-31

**Soundness:** 4
**Presentation:** 3
**Contribution:** 3
**Rating:** 8
**Confidence:** 4

**Summary:**

Inspired by the log-normal distribution of synaptic spine sizes in biological neural networks, this paper proposes the Log-normal Multiplicative Dynamics (LMD) algorithm for stable learning under low-precision computation.

The LMD algorithm is built on the Lie group Bayesian learning rule, giving rise to multiplicative weight updates which achieve large dynamic range at limited bit widths. The algorithm also incorporates multiplicative noise injections which stabilize weights for low-precision forward passes, as well as multiplicative weight decay which regularizes and prevents excessive weight growth.

Experiments with the Vision Transformer and GPT2 demonstrates state-of-the-art accuracies when performing low-precision forward passes using the MX data formats. This is good news for the design of energy-efficient hardware for inference.

**Strengths:**

**Significance**
* Promising results on vision transformer and GPT2 architectures

**Originality**
* Proper treatment of multiplicative noise injection
* Combination of many useful ideas (signSGD, signed weights trick, decoupling of gradient and regularizer in the momentum, multiplicative weight updates, multiplicative weight decay, multiplicative noise injection) and showing that they actually work together well.

**Clarity**
* Simple clear implementation in PyTorch for existing models, via drop-in replacement of the usual Adam optimizer, and a simple adaptation of the existing parameter initialization scheme.

**Quality**
* High quality ablation studies show effects on accuracy, regularization and low-precision forward passes that come from multiplicative weight updates, weight decay and noise injection in the learning algorithm.

**Weaknesses:**

1. Hard to understand LMD Algorithm 1 without first understanding Lie Group BLR. One should assume that the reader is not familiar with LGBLR. It takes a while to see that Eq 5 and Algorithm 1 are largely the same, after features like signSGD and gradient/regularization separation are removed. For instance, how does one decide to add signSGD to v_temp and not to the regularizer R? Perhaps all this can be explained in more detail in the Supplementary Material, expanding the discussion in Section 2.3. After all, this paper is ultimately about the LMD Algorithm.

2. Scaling the gradient by the weight is only briefly mentioned in Eq 5, Section 2.3 and Section 4.2. Does this scaling come from Lie Group BLR? Please do explain this a little if you are going to talk about its effect on regularization. Intuitively, how does it arise naturally from multiplicative Lie groups, but somehow not in classical MWU? If there is not enough space, this discussion could go in the Supplementary Material.

3. In Section 3.2 on Initialization, it took me a while to understand that initialized values are sampled using m\epsilon where \epsilon is logNormal(0, \sigma^2), and that you need to apply E[m\epsilon] = m \exp(\sigma^2/2) to find the mean of the initialized values. This could be made more explicit.

**Questions:**

1. How is it different from replacing every parameter m in a given model with \exp \mu, rederiving the gradient updates, applying weight decay and applying noise additively to \mu?

2. Any kind of indication how much slower it is to run MWU on GPUs for training large neural networks, than running additive updates on GPUs?

3. Why then did the sign function in Eq 1 disappear in Eq 5? Is it because the signs of the weights are preserved in LGBLR?

---

> ### Author Response · Authors · 2025-11-25
>
> We thank the reviewer for their very careful review and for pointing out that (i) the combination of many useful ideas necessary for the practical application of LMD, and (ii) high quality ablation studies. Above all, we are pleased that they provided suggestions to make the paper easier to understand.
>
> We have responded to the concerns raised below.
>
> > Comment 1 “Hard to understand LMD Algorithm 1 without first understanding Lie Group BLR.,..., For instance, how does one decide to add signSGD to v_temp and not to the regularizer R?”
>
> We updated Algorithm 1 to highlight in red the key differences from Eq. (5). We apply the regularizer R directly (without a sign) following the same strategy as Lion, and R includes a soft clipping term that applies a penalty of magnitude 1 when θ=1. This avoids performance degradation due to a scale mismatch between the gradient and the regularizer (added in L317–318). We also updated the title of Sec. 2.3 to include “via Lie Group Update” to clarify that it provides an overview of LGBLR.
>
>
> > Comment 2 “Scaling the gradient by the weight is only briefly mentioned in Eq 5, Section 2.3 and Section 4.2. Does this scaling come from Lie Group BLR?, …, Intuitively, how does it arise naturally from multiplicative Lie groups, but somehow not in classical MWU?”
>
> Thank you for your important comment. We have added the reason why (Multiplicative) LGBLR handles the scaling of the gradient by the weight to the L220-227. The main difference is whether the weight perturbation when evaluating the gradient is treated additively (classical MWUs) or multiplicatively (LGBLR).
>
> > Comment 3 “In Section 3.2 on Initialization, it took me a while to understand that initialized values, …”
>
> Thank you for your comment. We have made the initialization process clearer in the L372-373.
>
> > Q1 “How is it different from replacing every parameter m in a given model with \exp \mu, rederiving the gradient updates, applying weight decay and applying noise additively to \mu?”
>
> There is essentially no difference between your interpretation and our method. For gradient update and weight decay, we find agreement by applying m = exp(μ) to equation (10).
> For noise injection, adding Gaussian noise z to μ is equivalent to multiplicative lognormal noise ε, since ε = exp⁡(z) and exp⁡(μ)ε = exp⁡(μ+z). These relationships follow directly from the LGBLR formulation. In contrast, Kiral et al. do not discuss or actually use multiplicative weight decay. In their experiments, they employ standard additive weight decay with a Gaussian prior.
> > Q2 ”Any kind of indication how much slower it is to run MWU on GPUs for training large neural networks, than running additive updates on GPUs?”
>
> There is almost no reason why MWUs would be slower than additive update. We added the execution times for ViT and GPT-2 in Table 3, 4 (Appendix B) . LMD on bfloat16 is about 11% to 20% slower than AdamW on bfloat16, but this is mainly due to the overhead of weight sampling.
>
> > Q3 ”Why then did the sign function in Eq 1 disappear in Eq 5?”
>
> Thank you for pointing this out. We have corrected the sign in Eq. (5) (L208). Note that for Algorithm 1, matrix A plays the role of sing(m), so it is not necessary for Eq. (6).

---

### Official Review · Reviewer_C5eL · 2025-10-31

**Soundness:** 4
**Presentation:** 3
**Contribution:** 4
**Rating:** 6
**Confidence:** 3

**Summary:**

This work proposes parameter update rule based on multiplicative updates with log-normal noise. Prior works have attempted one of these two contributions but not both. The authors show that the new rule (LMD) enables stable low-precision training.

**Strengths:**

1. The paper's contributions enable low-precision (and therefore, low energy) training of large neural network which is timely and useful.
2. The paper's claims are well laid out and supported.

**Weaknesses:**

1. Evaluation is limited to only two model + dataset pairs.
2. Evaluation is limited to standard optimizers without focusing on low-precision schemes.

**Questions:**

Q1: How do you compare against other low-precision training schemes (e.g., [1] or [2])?

Q2: LMD seems to have overall lower training performance in Table 1. Why is that? Are there practical limitations to its convergence?

Q3: Could you estimate the potential compute savings? I know you cannot measure them due to emulation.

[1] : https://arxiv.org/abs/1803.03383

[2] : https://arxiv.org/abs/1710.03740

---

> ### Author Response · Authors · 2025-11-25
>
> We thank the reviewer for their review and for pointing out that (i) LMD effectively utilizes multiplicative updates and multiplicative noise, and (ii) LMD maintains stability even in low-precision forward learning. These properties, combined with multiplicative weight decay, are what allow us to overcome the scalability limitations of MWU.
>
>
> We have responded to the concerns raised below.
>
> > Comment 1: “Evaluation is limited to only two model + dataset pairs.”
>
> Based on feedback from other reviewers, we have added evaluations on small networks in Appendix C. In both cases, LMD shows superior results.
>
> > Comment 2: ”Evaluation is limited to standard optimizers without focusing on low-precision schemes.”
>
> In this study, we show that (i) in the bfloat16 setting, LMD achieves performance comparable to AdamW on GPT-2 and outperforms AdamW on ViT. We also confirm that (ii) when using low-precision forward, LMD exhibits almost no performance degradation compared to the bfloat16 baseline.
> This paper generally claims that when AdamW or LMD are used as the baseline (bfloat16), there is no performance degradation compared to these baselines, and this claim remains unchanged by results from other low-precision schemes.
>
> > Q1: “How do you compare against other low-precision training schemes (e.g., [1] or [2])?”, ”Evaluation is limited to standard optimizers without focusing on low-precision schemes.”
>
> Regarding [1], their study focuses on reducing the precision of the vectors used for parameter updates and does not consider low-precision matrix multiplication as in this paper, so their setup is clearly different. They perform matrix multiplication in full precision, i.e., FP32.
>
> The study in [2] discusses how to prevent performance degradation when replacing FP32 matrix multiplication with FP16, and their claim that there is no loss in model performance compared to the baseline is certainly similar to ours. We also cite this work in our paper [Micikevicius et al., 2018]. However, as already pointed out in the abstract of [Kalamkar et al., 2019], this approach requires careful tuning of hyperparameters. Instead, we evaluate LMD using bfloat16, the current standard for low-precision training, as a baseline.
>
>
> >Q2: ”LMD seems to have overall lower training performance in Table 1. Why is that? Are there practical limitations to its convergence?”
>
> In LMD training, multiplicative noise is injected into the weights. As a result, the model is less likely to overfit to the training data and the loss takes on relatively large values. In fact, multiplicative noise injection is considered a form of data augmentation [Trinh et al., 2024].
>
> Note that test performance of LMD is superior to other methods. With LMD, a low training score does not necessarily mean a low test score.
>
> > Q3: “Could you estimate the potential compute savings? I know you cannot measure them due to emulation.”
>
> In this paper, we focus on the stability of LMD as an optimizer. At the same time, as suggested in the reviewer's comments, an important motivation for us is to contribute to the compute savings that will be realized by future hardware.
>
> At this stage, our only results without performance degradation are for low-precision forward passes using MXFP6. Assuming NVIDIA GB300 Tensor Cores, (MX)FP6 has twice the peak performance of bfloat16, so ideally we can expect up to a 2× speedup in the forward pass.
>
> However, end-to-end training speedup also requires low-precision backward passes, which cost roughly twice as much compute as the forward pass. While there are several promising methods for low-precision backward passes, low-precision forward passes remain relatively underexplored. We have added this discussion to L492–499.
>
> Furthermore, if future hardware can be designed specifically for MXFP6, it is estimated that MXFP6-based computing systems will have a hardware cost advantage of approximately 2x compared to FP8-based computing systems [Rouhani et al., 2023b]. This point has also been added to the L291–293.

---

### Author Response · Authors · 2025-11-25

We apologize for the delayed response. We thank the reviewers for their helpful comments.

Our source code is publicly available at the following URL:

https://anonymous.4open.science/r/FyBaTn2fmj9FUMdZdbJ2zyKXA9uh

This includes code and procedures for reproducing our results.

---

### Meta-Review · Area_Chair_D2VX · 2025-12-19

**Summary:**

This paper proposes Log-normal Multiplicative Dynamics (LMD), a novel parameter update rule combining multiplicative updates with log-normal noise to enable stable low-precision training of large neural networks (ViT, GPT-2). Reviewers acknowledge the timely contribution and strong experimental results. The primary concerns centered on the scope of empirical evaluation (limited models/datasets, lack of comparisons to other low-precision schemes), clarity of the algorithm's derivation and relation to prior work (Lie-group Bayesian learning), and practical utility assessments (computational cost/speed, ablation studies).

**Reviewer Concerns:**

The concerns that have partially addressed:

1.Empirical Scope (C5eL, YL91): Authors added experiments on smaller networks and provided runtime comparisons.

2.Clarity & Derivation (JeNJ): The authors improved the presentation of Algorithm 1 and added explanatory text.

3.Baseline Comparisons (YL91): Added a comparison to the standard MWU method (Madam) on smaller models.

Outstanding Core Concern (Innovation/Novelty):

1.Lack of Clear Differentiation & Theoretical Foundation (vFuX, C5eL): This remains the most significant flaw.

2.Integration vs. Innovation: The authors' rebuttal frames LMD as a successful combination of existing ideas (Lie-group updates, signSGD, multiplicative noise/decay) for scalability. However, they fail to establish a paradigm-shifting novel concept or a clear theoretical advance that justifies a high-impact publication. The comparison to the prior Lie-group method (Kiral et al.) in the appendix shows practical implementation differences but does not demonstrate a fundamental algorithmic or theoretical breakthrough.

3. The work is perceived as an incremental engineering improvement—making an existing theoretical framework (LGBLR) work reliably on large models by incorporating several known tricks. The central claim of enabling low-precision forward passes, while practically useful, is not tied to a novel property of LMD itself but rather to the inherent stability of multiplicative updates when combined with specific noise and regularization.

4.The reviewer's request for "theoretical support for why LMD achieves higher accuracy" was not met, with the authors stating it was not the paper's focus. This absence leaves the performance improvements as empirical observations without a compelling new conceptual understanding.

**Reviewer Scores:**

Reviewer C5eL (Initial: 6): While pleased with expanded evaluations, this reviewer's initial rating ("marginally above acceptance") already indicated reservations. The unresolved question about fundamental novelty and comparison to other low-precision schemes aligns with their score ceiling. Final score likely remains at 6.

Reviewer JeNJ (Initial: 8): This reviewer focused on clarity and was positive. They might maintain a high score based on technical soundness and improved writing. However, in a discussion highlighting the pervasive novelty concern from other reviewers, they might acknowledge the issue. Final score might slightly lower to 7.

Reviewer YL91 (Initial: 4): Their concerns about ablation and speed were addressed, potentially raising their score. However, the core "contribution: poor" assessment relates to incremental impact. Final score might rise to 5 or 6, but not to a clear accept.

Reviewer vFuX (Initial: 2): This reviewer explicitly cited lack of novelty and clear differentiation as reasons for rejection. The authors' response provided a practical comparison but did not address the fundamental critique of insufficient conceptual advancement. This reviewer would almost certainly maintain a firm score of 2 (Reject).

---

### Decision · Program_Chairs · 2026-01-26

Reject